Mental health; community engagement; decolonization; developing countries; schizophrenia

**Corresponding author:**
Ana Carolina Florence;
Email: ana.florence@nyspi.columbia.edu

# State of the art of participatory and user-led research in mental health in Brazil: A scoping review

Ana Carolina Florence[1,2] , Mateus Bocalini[3] , Daniela Cabrini[3] , Rita Tanzi[3] , Melissa Funaro[4] , Gerald Jordan[5] , Larry Davidson[6] , Robert Drake[1], Cristian Montenegro[7] and Silvio Yasui[3]

[1]Department of Psychiatry, Columbia University, New York, NY, USA; [2]New York State Psychiatric Institute, New York, NY, USA; [3]Department of Social Psychology, State University of São Paulo (UNESP/Assis), Assis, Brazil; [4]Harvey Cushing/John Hay Whitney Medical Library, Yale University, New Haven, CT, USA; [5]School of Psychology, Institute for Mental Health, University of Birmingham, Birmingham, UK; [6]Yale Program for Recovery and Community Health, New Haven, CT, USA and [7]Wellcome Centre for Cultures and Environments of Health, University of Exeter, Exeter, UK

## Abstract

Participatory research denotes the engagement and meaningful involvement of the community of interest across multiple stages of investigation, from design to data collection, analysis, and publication. Traditionally, people with first-hand experience of psychiatric diagnoses, service users, and those living with a psychosocial disability have been seen objects rather than agents of research and knowledge production, despite the ethical and practical benefits of their involvement. The state of the art of knowledge about participatory research in mental health Brazil is poorly understood outside of its local context. The purpose of this article was to conduct a scoping review of participatory and user-led research in mental health in Brazil. We identified 20 articles that met eligibility criteria. Participation in research was not treated as separate from participation in shaping mental health policy, driving care, or the broader right to fully participate in societal life and enjoy social and civil rights. Studies identified several obstacles to full participation, including the biomedical model, primacy of academic and scientific knowledge, and systemic barriers. Our extraction, charting, and synthesis yielded four themes: power, knowledge, autonomy, and empowerment. Participation in this context must address the intersecting vulnerabilities experienced by those who are both Brazilian and labeled as having a mental illness. Participatory research and Global South leadership must foreground local epistemologies that can contribute to the global debate about participation and mental health research.

## Impact statement

Our scoping review of participatory and user-led research in mental health in Brazil highlights the importance of engaging the community of interest in the research process and challenges the traditional view of people with psychiatric diagnoses as mere research objects. Our findings reveal that participatory research in mental health in Brazil is not treated as separate from participation in shaping mental health policy, driving care, or the broader right to fully participate in societal life and enjoy social and civil rights. We identified several obstacles to full participation, including the biomedical model, primacy of academic and scientific knowledge, and systemic barriers. By foregrounding local epistemologies and promoting Global South leadership in mental health research, our work contributes to the global debate about participation and mental health research. Our findings have implications for mental health research in Brazil and beyond, and we anticipate that our research will be used to inform the development of more inclusive and equitable research practices in mental health worldwide.

## Introduction

Participatory research denotes the engagement and meaningful involvement of the community of interest in multiple stages of investigation, from design to data collection, analysis, and publication. In the 1970s, participatory action research (PAR), influenced by the work of Brazilian author Freire (1987), emerged in the social sciences to challenge the neutrality of science and address power asymmetries between academic and popular knowledge (Borda, 2008). PAR proposes a North–South encounter based on solidarity and mutual enrichment paying special attention to the colonial legacy of oppressed countries. According to de Sousa

Santos (2015), an understanding of the world beyond the western understanding of the world exists; we need cognitive justice to have global social justice; and emancipation takes shape in diverse ways outside of Western theory (de Sousa Santos, 2015). As such, this type of research favors subject to subject interactions toward transformation, rather than subject to object, thus reshaping epistemological notions that underpin traditional research (Fals-Borda, 1987). Mental health research in the Global North incorporated principles of PAR to study problems *with* communities (Kidd et al., 2018), and a discrete body of knowledge led by people who identified as consumers, survivors, and ex-patients of psychiatry has developed (e.g., survivor research and mad studies) (Faulkner, 2017). The Alma Ata was the first international declaration to explicitly state that individuals have the right to participate in shaping healthcare (WHO, 1978). Since the 1980s, public and patient involvement (PPI) or community and public engagement, grown out of civil rights movements, governmental initiatives, and disability rights movements (Tomes, 2006; Sweeney et al., 2009), further consolidated the importance of participation in shaping public policy and in research (Hickey et al., 2022). This tradition continued to expand in high-income countries (HICs), and acceptance of the principles of participatory research by mainstream science is growing (Pearce, 2021). Examples of the institutionalization of PPI in HICs are the National Institute of Health Research funded INVOLVE in the United Kingdom (started in 1996), the Patient-Centered Outcomes Research Institute (PCORI) in the United States, and the 2009 International Collaboration for Participatory Health Research with all but one steering committee member coming from the Global North (Beresford and Russo, 2020).

Participatory research in mental health counters the history of a field defined by the politics of exclusion (Mills, 2014). Unlike other medical specialties, psychiatry's relationship to social order has shaped discourses (e.g., notions of the degenerative course of mental illness, mental illness as a moral problem, and mental illness as deficit), and interventions (e.g., long-term internment and involuntary hospitalizations) that resulted in the systematic exclusion of individuals from society (Foucault, 1999). While a shift from asylums as the privileged locus of "treatment" happened globally, and deinstitutionalization became a priority, the issue of social exclusion has not been resolved, and even as co-production and participation develop in research, power asymmetries remain (Rose and Kalathil, 2019). Research participation in low- and middle-income countries is further complicated by the enduring history of colonial legacies (Bulhan, 2015) and the geopolitics of knowledge (Naidu, 2021) that privileges Euro-American epistemologies (Mignolo, 2005), which materializes in science in multiple ways. These have multiple consequences: globally, more than 98% of all funding streams for mental health are awarded by HICs (Woelbert et al., 2020) who consequently occupy a privileged position with respect to knowledge production (Abimbola, 2019); representation of individuals who are white, male, and HIC-based working at academic places of power (e.g., editorial boards and universities) is disproportionate (Naidu, 2021). In addition, interventions developed in the Global North have been systematically adapted and implemented in the Global South, for example, the World Health Organization's mhGAP (Timimi, 2011). An equally consequential form of power and exclusion lies in the epistemological dominance of HICs, who determine the methods and knowledge that count, and continue to exclude, silence, and oppress forms of being and knowing originating in the Global South (Alejandro Leal, 2007; Bulhan, 2015; Bhakuni and Abimbola, 2021; Naidu, 2021). Psychiatry's

history of exclusion and marginalization combined with the epistemic violence that results from the geopolitics of knowledge that shape the North–South relationship places participatory research in mental health in the Global South at the intersection of multiple oppressions.

The psychiatric reform movement in Brazil took place in the context of broader societal changes toward democratization. Starting in the 1970s, mental health workers in the country organized to denounce the abuse and inefficiency of psychiatric hospitals to treat and support the recovery of people with severe mental health problems (Amarante, 1998). Inspired by Basaglia's Democratic Psychiatry and the experiences of deinstitutionalization in Italy, the anti-asylum movement grew side by side with Brazil's universal public health system, both informed by a strong critique of positivist and biomedical epistemologies as insufficient to address social problems (Yasui, 2010; Amarante, 2015). This paradigmatic shift from asylums to the psychosocial care system was enshrined in law in 2001 (Law 10.216). Service user participation is a key feature of the Brazilian public health system and built into the principles of the psychiatric reform movement as well as the public healthcare system. Despite numerous successful experiences of service user involvement and leadership in mental health policymaking, service delivery, and advocacy (Vasconcelos, 2009), effective and consistent participation remains aspirational.

Participation in research, however, is not as well established. More than 20 years since the shift in how mental health services are organized in Brazil has been enshrined in law, evaluation of mental health services using participatory methods remains scarce (Ricci et al., 2020). This historical context makes it so that participation in policymaking, advocacy, and research are not understood separately. The state of the art of mental health research in the country is unknown. This knowledge gap is problematic locally and globally. Locally, participatory initiatives remain isolated in the context of specific projects and a national agenda for the advancement of participation of service users and people with lived experience of mental health problems would benefit from this scientific knowledge base. Globally, researchers remain unaware of the wealth of knowledge produced in Brazil. Thus, our study sought to review the empirical participatory literature in mental health in Brazil, identify common themes, and synthesize the results.

## Objective

Our scoping review's objective was to chart and analyze the empirical participatory mental health research literature in Brazil. Our review focused on research studies and included gray literature. Specific objectives were to describe how participation is conceptualized in mental health research in Brazil, to identify key concepts associated with participatory research, and to identify the main obstacles to participatory research in mental health in the country.

## Methods

Our team included academics in Brazil, Chile, the United States, and the United Kingdom, and people with lived experience of mental health challenges. Our review followed the Joanna Briggs Institutes' guidelines (Peters et al., 2017) and PRISMA extension guidelines for scoping reviews (Tricco et al., 2018).

### Eligibility criteria

Eligibility included studies that used participatory research methods, broadly defined as research in which service users' and family members' roles in the study went beyond those of research subjects (i.e., an individual who provides information or data to help answer a research question). We defined participatory research procedures broadly and included studies that employed member-checking, consultations, co-production, data validation procedures, and stakeholder consultation groups. We included empirical studies conducted in Brazil and published in peer-reviewed journals in Portuguese, Spanish, English, or French (study team's languages). We included the gray literature as well. We did not specify dates. Studies using quantitative, qualitative, or mixed methods were eligible. Studies that included participants of any age, sex, gender, ethnicity, race, or class were included. Studies involved service users and/or family members with any mental health diagnosis but excluded those with a primary diagnosis of a physical health problem (e.g., epilepsy and dementia) or substance use exclusively. We excluded studies that claimed to be participatory but provided no evidence of participation in the methods or results section of the paper. We excluded studies focusing on providers only but kept those that included providers if families or service users were included as well.

### Information sources

Our initial exploration of the topic revealed a series of challenges to traditional search strategies. Key words and vocabulary were inconsistent, metadata were missing, and titles were not available in databases (e.g., Web of Science and Scopus). To address these challenges, our team developed a multipronged strategy that combined bidirectional citation tracking (Hinde and Spackman, 2015) and a targeted search strategy to a variety of databases to identify relevant studies. We conducted two bidirectional searches and one additional search of targeted databases.

The research team, through their knowledge about this literature, Google Scholar searches, and consultations with experts, first identified 10 relevant studies (known as "pearls"), and the medical librarian used these pearls to identify articles through a systematic search of their cited and citing articles using citationchaser, SciELO, Scopus, Web of Science, and Google Scholar.

Using the included studies, we identified relevant terms using the Systematic Review Accelerator WordFreq tool (Clark et al., 2020) and developed a targeted search strategy. The librarian searched the following databases: MEDLINE, Embase, PsycInfo, Global Health, LILACS, Web of Science, Scopus, SciELO, BDTD, and the PBiPortal de BuscaIntegrada. We limited search results to English and Portuguese titles because the previous step did not yield relevant results in French and Spanish.

Our team translated the search strategy to Portuguese between the following databases to find published and unpublished (i.e., gray) literature: PubMed, MEDLINE (Ovid), Embase (Ovid), PsycInfo (Ovid), Web of Science Core Collection, Scopus, and Scielo.br. We pooled results in EndNote, removed duplicates, and uploaded to Covidence. We identified relevant theses and dissertations using the National Thesis Database BDTD and relevant books using the Universidade de São Paulo library catalog. Finally, our team consulted experts in the field, charted the main publication venues outside mainstream academic databases (ABRASME, APRAPSO, and ABRASCO for conference proceedings), and hand-searched key journals (e.g., Revista de Saúde Coletiva).

### Selection of sources of evidence

Two independent reviewers screened studies' titles, abstracts, and full texts using screening checklists that were pilot-tested and adjusted using the first 100 articles. Decision trees helped resolve ambiguous situations during the screening process. Ultimately, the study team decided to exclude the gray literature because most did not present primary studies. Many studies did not describe the methods making it difficult to assess if they were primary studies or not. Relevant theses and dissertations that were empirical research generally had an associated peer-reviewed publication, which we included.

### Data charting process

Our team iteratively developed a data charting form using Covidence and Excel and used it to extract relevant information (Tables 1 and 2). The first author and a member of the study team extracted the data independently and resolved conflicts together. We consulted a third member of the study team when consensus could not be reached.

### Data items and synthesis of results

Synthesis of results was iterative. We extracted data items that were relevant to the objectives of our review first in Covidence and then in Excel, including: article information (i.e., title, authors, year of publication, and aims); demographic information (i.e., age, sex, gender, ethnicity, race, and socioeconomic status); clinical characteristics (i.e., mental health problems) of the sample; whether participants were service users, family members, or providers; the study setting (i.e., community mental health center, primary care, and university); the methodological and analytical approaches; and definitions of the participatory elements in the study (e.g., member checking, designing research questions, and co-production). We used Atlas.ti to free-code the articles, and finally inductively developed a set of categories by grouping and organizing the codes.

### Results

We identified 1,437 references through the search strategy. After removing 814 duplicates, we screened the titles and abstracts of 974 references. We assessed 536 full text studies for eligibility and excluded 516 for several reasons (Figure 1), leaving final pool of 20 studies to be charted and synthesized.

### Characteristics of sources of evidence

Study publication dates ranged from 2009 to 2021. Most studies were carried out in the South and Southeast areas of Brazil ($N = 13$) and published in Portuguese ($N = 19$). Sample sizes ranged from 7 to 420, with a median number of 15 participants. Most studies were conducted at Community Mental Health Centers ($N = 14$) and employed qualitative methods ($N = 18$). Studies used a variety of data analysis approaches, with hermeneutic analysis ($N = 7$) being the most common. Most studies included participants diagnosed with serious mental illness, psychosis, or both ($N = 18$). None of the studies reported full demographic characteristics (i.e., sex, age, and race/ethnicity).

**Table 1.** Demographic, clinical, and methodological characteristics of the included studies

| ID | References | Aim | Sample size | Age | Sample | Sex/gender | Ethnicity/ race | Socioeconomic status |
|---|---|---|---|---|---|---|---|---|
| 1 | Alves et al. (2018) | To build autonomy-related qualitative outcome measures for Psychosocial Care including the perspectives of service users and their families | N = 18 | N/A | Adults | Female = 9 Male = 9 | Black = 5 Mixed = 6 White = 16 | N/A |
| 2 | Chassot and da Silva (2018) | To support the development of a user-led association | N/A | N/A | Adults | N/A | N/A | N/A |
| 3 | Emerich et al. (2014) | To identify and understand how mental health service users and managers conceptualize service user rights | N/A | N/A | Adults | N/A | N/A | N/A |
| 4 | Garcia et al. (2017) | To discuss a training experience that brought together academics, graduate students, and mental health service users and providers in the Northeast region of Brazil | N/A | N/A | Adults | N/A | N/A | N/A |
| 5 | Gonçalves and Campos (2017) | To evaluate the uptake of the Medication Management Guide (GAM) by mental health service users when talking to their providers and in their political engagement | N = 7 | 31–50 years of age | Adults | Male = 7 | N/A | N/A |
| 6 | Jorge et al. (2012) | To analyze the experiences of mental health service users with the development of the Medication Management Guide (GAM) group | N = 13 | N/A | Adults | N/A | N/A | N/A |
| 7 | Kantorski et al. (2009) | To report on a mixed-methods evaluation program of the public mental health centers in the Southern region of Brazil | N = 205 | Managers = 25–51 years of age; providers = 26–50 years of age; service users = 42; family members = 49 | | N/A | N/A | Managers = 50% had a specialization Providers = 55% had college education; 40% had postgraduate education Service users = 91% know how to read; 54% completed secondary school Family members = 52% had not completed secondary school; 11% completed high school; 2% had a college degree |
| 8 | Lima et al. (2014) | To develop treatment outcome for the treatment of autism in the public mental health children and adolescent centers in Rio de Janeiro | Providers = 5–7 in each of the 14 focus groups conducted. Family members = 7–12 in each of the three focus groups. Total not reported. | N/A | Children | N/A | N/A | N/A |
| 9 | Massa and Moreira (2019) | To understand the views of people living in residential services in the state of Sao Paulo regarding health and healthcare | N = 10 | 24–85 years of age | Adults | Female = 5 Male = 5 | N/A | Secondary school = 2 Incomplete secondary school = 7 Illiteracy = 1 |

*(Continued)*

**Table 1.** (*Continued*)

| ID | References | Aim | Sample size | Age | Sample | Sex/gender | Ethnicity/race | Socioeconomic status |
|---|---|---|---|---|---|---|---|---|
| 10 | Moreira (2021) | To analyze the protagonism of people with serious mental illness in graduate-level health education | N = 58 | Students = 20; service users = 65 | Adults | N/A | N/A | N/A |
| 11 | Moreira and Onocko-Campos (2017) | To present the ways in which users of different psychosocial care centers perceive possible mental health actions in primary care based on the psychosocial care network | N = 12 | 30–66 years of age | Adults | N/A | N/A | N/A |
| 12 | Onocko Campos et al. (2009) | To analyze the assistance, management, and workers' education models of a network of psychosocial healthcare services (CAPS). | N = 420 | N/A | Adults | N/A | N/A | N/A |
| 13 | Onocko Campos et al. (2012) | To adapt and implement the Canadian Medication Management Guide translated to the Brazilian context and assess its use in mental health education and training | Four focus groups with 7–9 participants in each. Total not reported. | N/A | Adults | N/A | N/A | N/A |
| 14 | Palmeira et al. (2021) | To understand the experience of service users who attend peer support groups in a city in the state of Rio de Janeiro and assess how their attendance strengthened their ability to be protagonists in the Brazilian psychiatric reform | N = 9 | 34.6 (7.3) | Adults | N/A | N/A | N/A |
| 15 | Palombini et al. (2020) | To evaluate the Medication Management Guide in public mental health services in three regions of Rio Grande do Sul | N/A | N/A | Adults | N/A | N/A | N/A |
| 16 | Passos et al. (2020) | To discuss the innovative approach called *Support research* to democratizing services, and share experiences and knowledge between workers and promoting co-management | N/A | N/A | N/A | N/A | N/A | N/A |
| 17 | Senna and Azambuja (2019) | To report on an experience of mental health service users who led an education and training activity in a university in Rio Grande do Sul | N/A | N/A | N/A | N/A | N/A | N/A |
| 18 | Serpa Junior et al. (2014) | To investigate the meanings related to the experience of being diagnosed with schizophrenia from the perspective of service users and psychiatrists | N = 27 | Service users = 44; psychiatrists = 32 | Adults | Female = 12 Male = 15 | N/A | Service users: Incomplete secondary school = 4 Complete secondary school = 2 Incomplete high school = 1 Complete high school = 9 College degree = 1 No information = 1 Psychiatrists: Graduate degree = 7 Postgraduate degree = 2 |
| 19 | Silveira et al. (2014) | To report on a multicentric research project that included | N/A | N/A | Adults | N/A | N/A | N/A |

**Table 1.** (*Continued*)

| ID | References | Aim | Sample size | Age | Sample | Sex/gender | Ethnicity/ race | Socioeconomic status |
|----|-----------|-----|-------------|-----|--------|-----------|-----------------|----------------------|
| | | academics and mental health service users and providers | | | | | | |
| 20 | Vaz et al. (2019) | To reflect on the research knowledge committee of a national study that evaluated a social benefit for people with long history of institutionalization in 12 cities in Brazil | N/A | N/A | Adults | N/A | N/A | N/A |

| ID | Mental health problem | Participants | Setting | Overall methods | Epistemological tradition (not sure what to call this, theoretical foundations?) | Data collection | Data analysis |
|----|----------------------|--------------|---------|-----------------|------------------------------------------------------------------------------------|-----------------|---------------|
| 1 | SMI | Service users; family members; providers | Community Mental Health Center (CAPS) | Qualitative | Constructivism; hermeneutic dialectic | Interviews; focus groups; secondary data analysis | Thematic analysis |
| 2 | SMI | Service users | Service User Association | Qualitative | Community-based participatory research; participatory action research; psychoanalysis; institutional analysis | Interviews; participatory research group | Group discussion of the results involving researchers and participants |
| 3 | SMI; psychosis | Service users; family members; providers | Community Mental Health Center (CAPS) | Qualitative | Hermeneutic dialectic | Intervention groups; focus groups; interviews | Hermeneutic analysis |
| 4 | SMI | Service users; providers | University | Qualitative | Cartography; institutional analysis; intervention research | Cartographic mapping; field journals; group conversations | Institutional analysis |
| 5 | SMI | Service users | Community Mental Health Center (CAPS) | Qualitative | Hermeneutic dialectic | Interviews; focus groups; narrative groups | Triangulation of interviews and narratives built through focus groups; sequential independent fluctuating cross-reading of materials |
| 6 | SMI; substance use | Service users | Community Mental Health Center (CAPS) | Qualitative | Hermeneutic | Focus groups; narrative groups | Hermeneutic analysis |
| 7 | N/A | Service users; family members; providers | Community Mental Health Center (CAPS) | Mixed methods | Constructivism; hermeneutic dialectic | Interviews; participant observation; member checking | Hermeneutic analysis |
| 8 | SMI | Family members; providers | Community Mental Health Center (CAPS) | Qualitative | Hermeneutic | Focus groups; member checking | Thematic analysis |
| 9 | SMI; psychosis | Service users | Residential facility | Qualitative | Phenomenology | Focus groups; narrative groups | Development of themes; validation of results |
| 10 | SMI | Service users; family members | Community Mental Health Center (CAPS); others | Qualitative | Popular education | Interviews; focus groups | Thematic analysis |
| 11 | SMI; psychosis | Service users | Community Mental Health Center (CAPS) | Qualitative | Hermeneutic | Focus groups | Hermeneutic analysis |
| 12 | SMI | Service users; family members; providers | Community Mental Health Center (CAPS) | Qualitative | Hermeneutic dialectic | Focus groups; narrative groups; member checking | Hermeneutic analysis |

(*Continued*)

**Table 1.** (*Continued*)

| ID | Mental health problem | Participants | Setting | Overall methods | Epistemological tradition (not sure what to call this, theoretical foundations?) | Data collection | Data analysis |
|---|---|---|---|---|---|---|---|
| 13 | SMI; psychosis | Service users; family members; providers | Community Mental Health Center (CAPS); others | Qualitative | Hermeneutic | Focus groups; narrative groups; member checking; others | Hermeneutic analysis |
| 14 | SMI; psychosis | Service users | University | Mixed methods | Phenomenology | Interviews; member checking | Phenomenological analysis (qualitative); *t*-test (quantitative) |
| 15 | SMI | Service users; family members; providers | Community Mental Health Center (CAPS); primary care; specialized care; social work services | Qualitative | Paideia method | Narrative groups; group conversations | Hermeneutic analysis |
| 16 | SMI | Service users; family members | Community Mental Health Center (CAPS) | Qualitative | Intervention research; support research; institutional analysis | Autonomous medication management groups | N/A |
| 17 | N/A | Service users | University | Qualitative | Fourth-generation evaluation; paideia method; Foucault's archaeogenealogy | Participant observation | Foucault's archaeogenealogical perspective |
| 18 | SMI; psychosis | Service users; providers | Community Mental Health Center (CAPS) | Qualitative | Phenomenology; medical anthropology | Focus groups | Interpretive phenomenological analysis |
| 19 | SMI; psychosis | Service users; providers | Community Mental Health Center (CAPS); university | Qualitative | ResearchWITH | Joint writing | N/A |
| 20 | SMI; psychosis | Service users; providers; others | | Qualitative | Participatory research; methodologic triangulation | Group conversations | Triangulation |

Nine studies reported on a multicentric study in partnership with a Canadian university that translated and implemented a medication management guide in community mental health centers in several regions of Brazil (Jorge et al., 2012; Onocko Campos et al., 2012; Emerich et al., 2014; Silveira et al., 2014; Gonçalves and Campos, 2017; Chassot and da Silva, 2018; Senna and Azambuja, 2019; Palombini et al., 2020; Passos et al., 2020). Seven studies reported using principles of Guba and Licoln's fourth-generation evaluation, a constructivist method – focused on negotiation – that engages stakeholder groups across all phases in iterative processes to reach consensus (Guba and Lincoln, 1989, 2001; Kantorski et al., 2009; Onocko Campos et al., 2009; Jorge et al., 2012; Emerich et al., 2014; Moreira and Onocko-Campos, 2017; Alves et al., 2018; Palombini et al., 2020).

Group validation of results was the most common participation strategy (Kantorski et al., 2009; Onocko Campos et al., 2009; Jorge et al., 2012; Emerich et al., 2014; Gonçalves and Campos, 2017; Alves et al., 2018; Massa and Moreira, 2019; Palmeira et al., 2021). Only two studies explicitly stated that participants were involved in all stages of research (Chassot and da Silva, 2018; Vaz et al., 2019). Participants were not co-authors in any of the included studies, nor was there mention of authors' lived experience in any of the studies we included. Descriptions of the

value and role of participation in research varied. Authors acknowledged the importance of community participation in public policy as a means to connect research and action (Lima et al., 2014), noting that participation in research addresses power imbalances in the researcher–subject dyad (Passos et al., 2020). Another study marked diversity, respect and differences, and acknowledgment of lived experiences as legitimate sources of expertise as important reasons for participation (Onocko Campos et al., 2009). One study described participation as a tool to increase political reflection, bring attention to the rights participants may have lost, and increase the relevance of research (Moreira, 2021). Studies that employed fourth-generation evaluation methods highlighted the importance of stakeholder involvement in all stages of research to level power asymmetries in research and increase the relevance of knowledge produced (Jorge et al., 2012; Emerich et al., 2014; Alves et al., 2018; Senna and Azambuja, 2019; Palombini et al., 2020). One example of fourth-generation evaluation included service users, managers, psychiatry residents, and family members to translate, adapt, and test a medication management tool for people with serious mental illness. Their inclusion fostered a sense of agency in the research process, and researchers see themselves as social actors sharing the experience of the world and bringing their

**Table 2.** Participation definition, operationalization, co-authorship

| ID | References | Participatory process/definition | How did people participate? | Were participants co-authors? |
|---|---|---|---|---|
| 1 | Alves et al. (2018) | Fourth-generation evaluation – importance of stakeholder engagement. | Validation of results in focus group sessions. | No |
| 2 | Chassot and da Silva (2018) | Participatory intervention research: a confluence of influences including Brazilian health service stakeholder involvement tradition, health and mental health participatory research, and intervention research. | Participants were co-researchers and participated in all stages. | No |
| 3 | Emerich et al. (2014) | Fourth-generation evaluation – importance of stakeholder engagement. | Narrative validation through hermeneutic focus groups. | No |
| 4 | Garcia et al. (2017) | Problematizing the relationship between academic and popular/community knowledge in the public health field. Leveling the dialogue across disciplines. | Attending groups at the university | Unclear |
| 5 | Gonçalves and Campos (2017) | Use of narratives as a means to access experience, not turning voices into objects, researching with and not about. | Narrative validation through hermeneutic focus groups. | No |
| 6 | Jorge et al. (2012) | Centering the lived experience of participants of the Autonomous Management Groups with an emphasis on the experiences between service users and their provider team; fourth-generation evaluation. | Narrative validation through hermeneutic focus groups. | No |
| 7 | Kantorski et al. (2009) | Qualitative evaluation participatory research, supported by the Gadamerian hermeneutics. | Narrative validation through hermeneutic focus groups. | No |
| 8 | Lima et al. (2014) | Partnership between researchers and providers and family members. Highlighting the importance of participation in the process of public policy formulation and the relationship between research and action. | Workshop to jointly build the final measures. | No |
| 9 | Massa and Moreira (2019) | To look for and respect the meanings that participants attribute to the studied phenomena and understanding research as a way to produce knowledge. | Narrative validation through hermeneutic focus groups. | No |
| 10 | Moreira (2021) | Horizontal and reciprocal relationship in the production of knowledge by using collaborative approaches. Multiple types of interventions (poetry, teaching, and learning) connecting the mental health field with the struggle for human rights. | Weekly community meetings and workshops. | No |
| 11 | Moreira and Onocko-Campos (2017) | Participatory research as collective knowledge, creating ways for people to participate in the right and power to think, produce and direct the uses of their knowledge about themselves. A type of epistemological anticolonialism. | Focus groups to discuss all stages of research. | No |
| 12 | Onocko Campos et al. (2009) | Qualitative evaluation participatory research, supported by Gadamerian hermeneutics. | Participants elected the key problems to be addressed; narrative validation through hermeneutic focus groups. | No |
| 13 | Onocko Campos et al. (2012) | Hermeneutic focus groups based on Paul Ricoeur's perspective about narrative. | Participants were invited to research meetings and were part of the cultural adaptation of the guide | No |
| 14 | Palmeira et al. (2021) | Presupposes that participants know the experience and researchers will learn from them. | Validation of themes by study participants. | No |
| 15 | Palombini et al. (2020) | Fourth-generation evaluation and participatory support research. | Collective iterative analysis of narratives. | No |
| 16 | Passos et al. (2020) | Support research stimulates care and participation in the research process. | Co-leading intervention groups during the implementation process. | No |
| 17 | Senna and Azambuja (2019) | Balancing power differentials between researchers and participants. Research with and not about participants. | Participants were lecturers at the university. | No |
| 18 | Serpa Junior et al. (2014) | Using narratives to access the subjective experience and biography of participants. Service users learning about how psychiatrists think and vice versa. | Participants validated each other's narratives through focus groups. | No |
| 19 | Silveira et al. (2014) | Balance the distribution of expertise so that knowledge is not exclusively with the researcher. In this perspective, research actively involves everyone in a transformation process. | Service users wrote about the research experience. | No |
| 20 | Vaz et al. (2019) | Strengthen service user protagonism and increase participation of non-traditional agents in jointly creating dialogic knowledge. Overcome the subject–object dichotomy and its assumed scientific objectivity. | Stakeholders were invited to participate at the design, data collection, data analysis, and recommendation stages. | No |

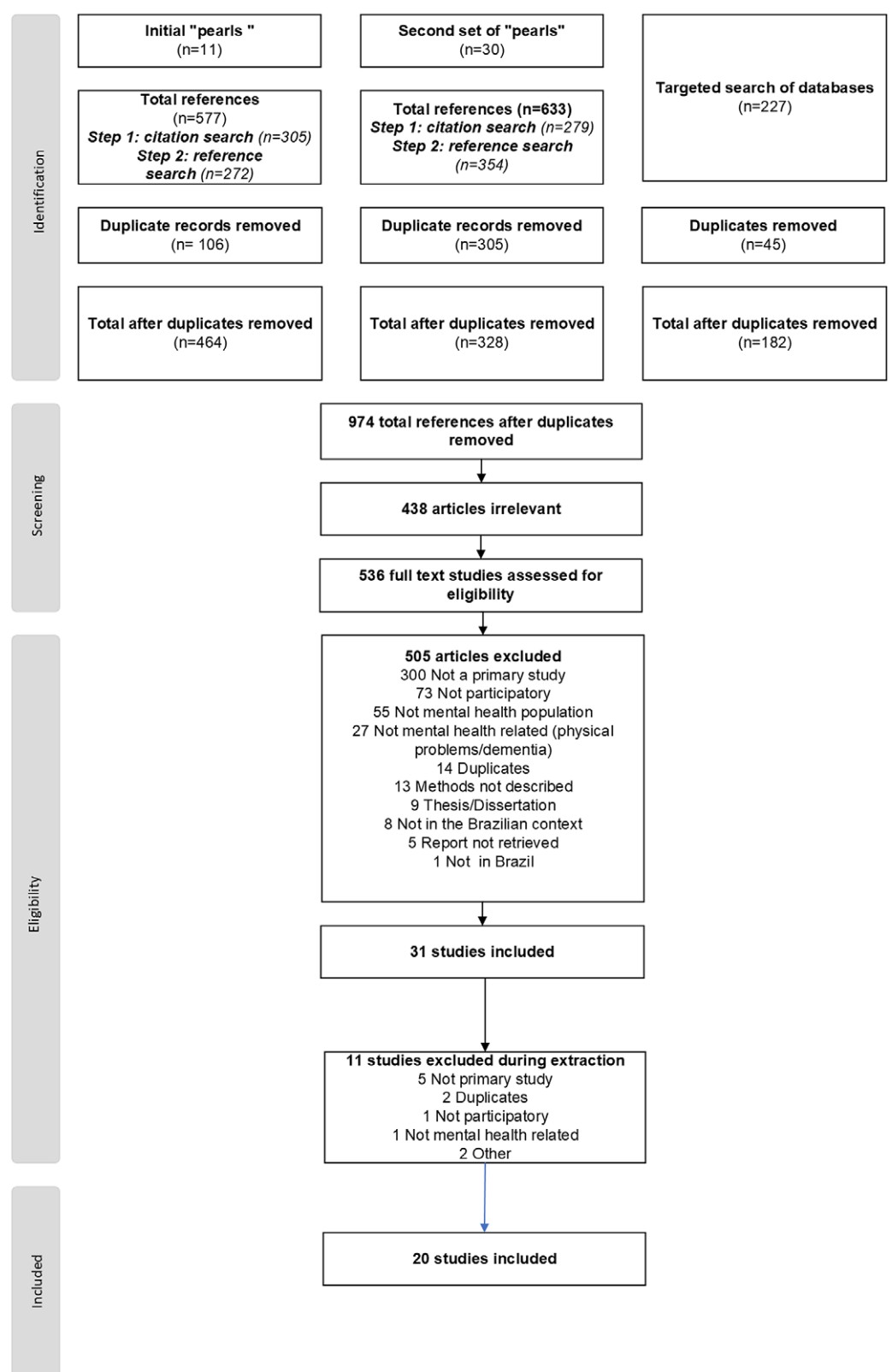

**Figure 1.** Screening and selection of articles.

own subjective experiences (Emerich et al., 2014). One study noted the need to increase participation in research, especially in mental health, given that, historically, service users have been excluded from decision spaces including about their own treatment (Gonçalves and Campos, 2017). One study noted that definitions and operationalization of participation vary greatly and can have different meanings (Moreira and Onocko-Campos, 2017).

## Synthesis of results

### Importance of participation

In the context of the Brazilian psychiatric reform movement, several studies have considered participation from various dimensions: policy, political, and clinical. From a policy perspective, studies note the shift from the biomedical, hospital-centric model of care to the creation of community-based mental health centers (CAPS) as the main locus of treatment in the public mental health system (Serpa Junior et al., 2014). From a political standpoint, participation in research and in shaping services emblematizes autonomy, citizenship, and the general exercise of civil liberties. Finally, the clinical perspective, more directly related to treatment encounters and service delivery, amplifies the political by connecting suffering with exclusion and marginalization; and treatment with freedom, autonomy, and participation in society.

> It [action research] is an essentially political way of doing research, which aims at the promotion of citizenship and focuses on the processes of social exclusion. (Moreira and Onocko-Campos, 2017, p. 465)

### Power and knowledge

A prevailing theme among selected articles is that psychiatry has historically oppressed and silenced the people it aims to serve by placing excessive emphasis on scientific and professional knowledge, and by reducing individuals to their diagnoses and symptoms (Jorge et al., 2012; Emerich et al., 2014; Gonçalves and Campos, 2017; Moreira, 2021).

> Deemed incapable of living in society, subjects are silenced and their tragic experience, frustration, failure and everyday suffering are gradually removed from daily experience and turned into psychopathological categories. (Moreira, 2021, p. 1190, our translation)

Studies point to harmful diagnostic language and treatment practices, rooted in a reductionist biomedical model, that have harmed and violated the rights of people with mental health problems.

> The separation between knowledge and the experience of madness legitimized Psychiatry's knowledge supremacy and made interventions also an expression of a power-knowledge in the name of treatment. (Moreira, 2021, p. 1190, our translation)

Authors suggest a need to correct power imbalances as critical to advance mental healthcare. These perspectives are strongly rooted in the works of Foucault and Basaglia.

### Autonomy and empowerment

Empowerment has historical roots in the struggle for civil rights in Brazil starting in the 1970s. Grounded in the work of Freire (1987) and popular education, this tradition motivated the public health and mental health reform movements to transform traditional forms of power and knowledge and foreground the rights of the historically oppressed (Garcia et al., 2017). In mental health, authors note that empowerment can be paradoxical because the need for special social rights (e.g., benefits, protected work, and free transportation) often clashes with universalist claims of civil rights (e.g., equal rights, social inclusion, and full participation in society) due to the extreme disenfranchisement of populations with intersecting vulnerabilities (e.g., extreme poverty, psychiatric diagnosis, violence, and food insecurity) (Gonçalves and Campos, 2017).

> In a country in which the precarity of access to social rights for survival is constant, the service user in intense psychic suffering seems to, often, experience a double process of exclusion: to be Brazilian and to be mad. (Emerich et al., 2014, p. 686, our translation)

In clinical care, lack of empowerment means not having enough information to make decisions about treatment. This is reinforced by power imbalances favoring professional and academic knowledge (Onocko Campos et al., 2012). Increasing autonomy is an important treatment outcome in the selected studies (Kantorski et al., 2009; Onocko Campos et al., 2009; Jorge et al., 2012; Emerich et al., 2014; Lima et al., 2014; Gonçalves and Campos, 2017; Alves et al., 2018; Chassot and da Silva, 2018; Senna and Azambuja, 2019; Palombini et al., 2020; Passos et al., 2020; Moreira, 2021; Palmeira et al., 2021).

> Historically, the Psychiatric Reform movement posed the redefinition of the meaning of autonomy for community based mental health service users as a clinical-political challenge. This meaning of autonomy must broaden and even shift the meaning inaugurated by modernity, because autonomy is no longer conceived as strictly individual. In the Brazilian Psychiatric Reform movement, the process of becoming autonomous and of emancipation is considered collective and shared. (Gonçalves and Campos, 2017, p. 1545, our translation)

### Obstacles to full participation

Participation in research, treatment, and societal life were intertwined in most studies and not analyzed separately. Key obstacles to full participation were conceptions of mental health (Jorge et al., 2012; Serpa Junior et al., 2014) (e.g., the biomedical model and psychopathology); systemic issues (Emerich et al., 2014; Palmeira et al., 2021) (e.g., tutelage, violence, poverty, and lack of access to healthcare and basic rights); power asymmetries (Onocko Campos et al., 2012; Gonçalves and Campos, 2017; Moreira and Onocko-Campos, 2017) (e.g., primacy of academic and professional knowledge, infantilizing service users, and disenfranchisement in treatment).

> Consequently, the team's actions often are directed to the need for symptomatic remission, and their service users' words remain muted due to the consideration given to their symptoms. This suggests that the responses indicated by the teams are still supported by the medical-biological perspective of understanding the phenomena of mental suffering, which does not seem to match what should be the object of work in this new context: the existence-distress in relation with the social. (Moreira and Onocko-Campos, 2017, p. 471)

## Discussion

The overarching aim of this review was to chart and synthesize the participatory research in mental health in Brazil. We identified 20 relevant studies. Studies stressed the importance of participation in research as part of a broader democratizing process, reshaping power and knowledge relationships between expert and experiential knowledge. Studies noted that empowerment and autonomy are at the center of the Brazilian Psychiatric Reform movement and that participatory research lends itself to support this emancipatory project. Included studies highlighted that Brazilian mental health service users endure intersecting and synergistic processes of social exclusion that must be acknowledged. The biomedical model's reductionist views of mental health, violence, poverty, and social

exclusion were all identified as barriers to full participation in research and in shaping public policy.

Overall, Brazilian researchers did not define participatory research as distinct from other important participatory processes in society, including mental health service evaluation, public policy, advocacy, and broader claims of rights and liberties citizens must be entitled to. By refusing to treat these domains separately, Brazilian researchers have emphasized service evaluation and qualitative research in lieu of efficacy and effectiveness trials to establish the evidence base of specific interventions. This may be the result of the historical partnership of mental health professionals who consider themselves militants and advocates for the rights of service users. This configuration is not as common in the Global North, which tends to place service users and providers on opposite political sides with conflicting interests.

Brazilian participatory research has been gradually developing in the country, and included studies highlighted the importance of participation; however, most studies only included participants in member checking activities. Claims of full participation were not substantiated or well described, and authorship was limited to researchers, even when there were claims of a participatory writing process. None of the studies reported full demographic information (age, sex/gender, and race/ethnicity), suggesting that reporting practices are inconsistent throughout and are not specific to participation. Participation is further complicated by multiple vulnerabilities mental health service users experience in Brazil, including poverty, limited literacy, and social exclusion. The tension between special and universal rights, a perhaps false tension if we consider the United Nations Convention on the Rights of Persons with Disabilities (CRPD) and a rights-based approach to mental health (Pūras, 2017), featured in included studies. The emphasis of this literature on the concepts of autonomy and empowerment challenges the paternalistic nature of special benefits as they often rely on specific diagnoses and imply a deficit, perpetuating the narrow biomedical and individual-centered understanding of disabilities. On the other hand, special rights have historically played an important role in overcoming past and ongoing oppression and exclusion of this population promoting social justice.

We note that nearly half of the included studies were conducted in partnership with a Global North country, suggesting that native participatory experiences not mediated by HICs are even rarer. The existence of well-developed theories of participatory research in Brazil using local epistemologies suggests that the scarcity of this type of research may be more related to funding inequities than a deficit in the field.

Since 1988, participation in public policy development and broader political participation in Brazil has been expanding, especially in public health. In mental health, participatory service evaluation is one of the most developed areas in which participation has been more fully implemented (Ricci et al., 2020); nevertheless, projects still fail to center service users and often only include other stakeholders (Furtado et al., 2013) (e.g., policymakers, administrators, providers, and leadership). Compared to HICs, Brazil still lags in this area. This must be understood within the broader landscape of global mental health funding inequities and consider that 98% of all mental health research funding comes from HICs (Woelbert et al., 2020). A systematic review of the literature shows that mental health research in Brazil places a bigger focus on providers and work processes rather than outcomes and service users' perspectives (de Rosalmeida Dantas and Raimundo Oda, 2014). Although the Psychiatric Reform movement has promoted a shift in the traditional psychiatric paradigm, the biomedical model of psychiatry is still pervasive in community-based mental health services (Jorge et al., 2012; Serpa Junior et al., 2014); thus, power asymmetries in care continue, further supporting the need for increased participation. Influences from HICs appear in how participation is understood and practiced in mental health research in Brazil (notably in the use of fourth-generation evaluation methods); however, it is safe to say that the history of the Sanitary and Psychiatric reform movements in the country and the local scholarly traditions shaped a well-defined autochthonous field. Participation in Brazil does not fit neatly into conceptual definitions from the Global North, such as user-led (Rose, 2003) and co-creation (Greenhalgh et al., 2016); instead, Brazilian authors stress that participation has multiple meanings and applications. Importantly, participation is strongly rooted in what Brazilian authors call collective autonomy and in the rights of those who face multiple vulnerabilities and social exclusion (Passos et al., 2013). While tensions between research and activism are common (Montenegro, 2018), the neat separation of interests between providers and service users is not relevant in the Brazilian context, and partnerships across stakeholders are fundamental.

It is noteworthy that 9 out of the 20 selected studies reported on a partnership with a Canadian university. Globally, mental health funding is unevenly distributed with HICs setting the agenda for the rest of the world. Along with setting the agenda, partnerships between the Global North and Global South in research are likely to be asymmetric despite good intentions. The project that yielded almost half of the publications we report here involved the translation and adaptation of a Canadian instrument, and not the creation of a native instrument. Such partnerships rarely ensure sustainability, and the impact of such projects tends to be limited to the life of the grant. Our study team has had multiple experiences with progressive researchers from the Global North who refuse to acknowledge a partnership between mental health providers and service users is possible. By setting the terms of what participation means, and what counts and does not, the Global North effectively continues to erase our history and deny our entrance in the debate unless we do so from a place of need and helplessness.

## Limitations

Our study had limitations. Our group had to build creative strategies to overcome challenges related to missing metadata and unorthodox scientific reporting practices. The definitions of empirical research do not fully map out to how research is conducted in Brazil, making it difficult to extract data using traditional methods. Our team excluded theses and numerous articles because it was not clear if they were empirical studies or not, even though some presented data. Studies using cartographic methods were largely excluded, though many claimed to be participatory. While reviews of the scientific literature are important, they fail to include other forms of knowledge production and participation. This is particularly true for the Global South and relates to the inequities mentioned in our discussion.

## Conclusions

Our study reveals important knowledge gaps. Participatory procedures were generally not well described except for the narrative data validation focus groups which are clearly conceptualized and form a discrete body of literature with well-described methods and

procedures. During the screening process, this became evident to our group, and we were forced to exclude potentially relevant articles because methods were not well described or were not described at all. Given the importance of participation in mental health research, the field could benefit from more standardized ways of reporting participatory procedures, which would in turn create better accountability of what counts as meaningful participation. We must acknowledge that the scientific literature does not fully capture the wealth of knowledge production and participation in the country. New and creative methods to synthesize the scientific and non-scientific (e.g., artistic) knowledge base produced by mental health service users are needed.

Among the numerous obstacles to participation, disenfranchisement, poverty, and lack of access to social and civil rights are particularly relevant. The intersecting vulnerabilities of being Brazilian and diagnosed with mental illness substantively impact participation in research and beyond. Service users who rely on disability benefits to survive and advocate for universal civil rights exist in a paradox that cannot be sorted through research methodologies but forces us instead to contend with the political nature of research that is committed to social change.

Reparations for coloniality in global mental health research are due. Should the Global North be truly interested in leveling historical asymmetries and inequalities in the world, the first step should be the equitable distribution of research funding. HIC researchers must enter partnerships with the Global South from a place of curiosity and solidarity. Capacity building should be grounded in mutuality instead of technology/knowledge transfer. Participatory research in Brazil is rich and original. The challenges for its expansion and full implementation must be understood within a broader social context of disenfranchisement, poverty, and lack of fundamental rights. This field holds true to the Latin American origins of participation as a transformative and democratic exercise and to the tenets of the Psychiatric Reform movement that questioned the primacy of the biomedical model in mental health, exposed its contradictions, and paved the way to a community-based network of services that centered dignity and freedom as inalienable rights of every citizen. We hope to see participatory research in Brazil expand and flourish as it has in HICs, the rhizomes already exist.

**Open peer review.** To view the open peer review materials for this article, please visit http://doi.org/10.1017/gmh.2023.12.

**Data availability statement.** Data availability is not applicable to this review article as no new data were created or analyzed in this study.

**Acknowledgements.** This project is affiliated with and contributes to the Lancet Psychiatry Commission on Psychoses in Global Context.

**Author contribution.** A.C.F. had the original idea for the study and presented to all co-authors. G.J. assisted in the protocol writing and advised the team throughout. M.F. developed the search strategy and led the data management in EndNote and Covidence. D.C., R.T., and M.B. screened all title and abstracts and full texts along with A.C.F. A.C.F. and M.B. extracted the data. A.C.F. wrote the manuscript. All co-authors reviewed several iterations of the manuscript and approved the final version.

**Financial support.** This study was supported by the Foundation for Excellence in Mental Health Care.

**Competing interest.** The authors declare no competing interests exist.

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
