## [Reviewer Report]

December 20th, 2022

Dear Drs., Chibanda and Bass, 

We wish to submit an original review entitled “State of the art of participatory and user-led research in mental health in Brazil: a scoping review” for consideration by Global Mental Health.

We confirm that this work is original and has not been published elsewhere, nor is it currently under consideration for publication elsewhere.

In this paper, we report on participation of individuals with lived experience of mental health challenges in research in Brazil. This is significant because the state of the art of this field is largely unknown in South America, however there’s clear indication it is a rich and productive area of research with distinct characteristics when compared to how this area has developed in the Global North. 

Our review shows that participatory research in mental health in Brazil is strongly rooted in local scholarship. We present challenges to full participation that go beyond research. Brazilian service users face multiple and intersecting vulnerabilities that pose challenges to inclusion in society and in research alike. We believe this review can contribute to a more nuanced global debate about participation in Latin America.

Our team has no conflicts of interest to disclose. 

Please address all correspondence concerning this manuscript to me at ana.florence@nyspi.columbia.edu.

Thank you for your invitation to submit this manuscript. 

Sincerely,

Ana Carolina Florence, PhD

---

## [Reviewer Report]

*Comments to Author*: The article is highly relevant within the Brazilian mental health field scenario, as well as points out the complex challenges of participatory research in the country, despite having a strong tradition of participation in education and in the day to day health and mental health services. The article covers the most relevant literature of participatory research in the mental health field in Brazil, and also discusses the several possible reasons and determinants for the huge gap and differences between the Brazilian and North richer countries in relation to the development of participatory research in the mental health filed. The methodology used for literarture review is highly sophisticated and well applied during all the research procedures. However, it is necessary to consider that, beyond the scientific literature surveyed, the results summed up by the article does not mean that mental health service users in Brazil do not show other forms of autonomous ways of expression and literature production, like publications pf autobiographic and personal narratives of lives with their mental health problems, poetry, music lirics, sambas, arts, etc. Therefore, for a full evalluation of the level of empowerment and autonomy of the Brazilian service users, other forms of expressions should be also taken on account and surveyed,

---

## [Reviewer Report]

*Comments to Author*: This is overall an excellent piece that tackles a crucial and hitherto not well documented weakness with regard to user participation in mental health research in Brazil. It conveys the historical context well and summarizes the huge gaps that need to be addressed to fulfill the potential for research to be meaningful to all concerned. The authors went to considerable lengths to ensure that the regional publications and perspectives are represented here.

I have three minor suggestions for improvement, none of them difficult to do:

1. I think the following sentence is somewhat misleading and should be modified.

“Service user participation is a key feature of the Brazilian public health system, with numerous successful experiences …in mental health…”

This is true in principle (ie built into the principles of the reform) and sometimes but not generally in practice, based on what the authors themselves state later in the ms (as well as on my admittedly more limited experience). I would suggest making this distinction. If the authors disagree, they need to substantiate the claim that this is a key feature in practice across a broad range of Brazilian mental health services, and what they mean by successful user participation in that case.

2. The authors refer to “fourth generation evaluation” and provide a book reference that is only accessible for those who buy the book. Most readers will not be familiar with this term and they should give a brief summary of what it means, one or two sentences would be enough.

If possible, also cite an accessible example of such an evaluation.

3. It is true that some do perceive a paradox in receiving social benefits for disability (“special rights) and having full and equal social participation (universal rights). But not all do, since others believe that these kinds of rights are not inherently in conflict. It would be useful to spell out the issue in a bit more depth because many readers will not have a good understanding of the tensions. There would be no harm in stating their own position on this issue (although that is optional)

---

## [Reviewer Report]

*Comments to Author*: An extensive research network, directly involving different actors of mental health policies, in particular, the beneficiaries of public policies, was gathered for the development of the article. The theme of the participation of people with direct experience of psychiatric diagnoses was contextualized in the world scenario and obstacles to the systematization and dissemination of knowledge in the sense of epistemological power relations were revealed. The method was clearly presented, as well as the resources used for review and analysis strategies. Important contributions were made to the field of mental health in the sense of valuing the knowledge of the people with experience of psychiatric diagnoses in research.

---

## [Reviewer Report]

Dear Dr. Bass,

We are resubmitting our manuscript entitled “State of the art of participatory and user-led research in mental health in Brazil: a scoping review” for your consideration after incorporating the comments and suggestions of the editor reviewers. We appreciate the constructive feedback provided by the reviewers, which helped us improve the quality of our manuscript.

As per the reviewers' recommendations, we have made several changes to our manuscript. Specifically, we have addressed the following points:

- Suggestions from reviewers 1 and 3

- Formatting issues

- Graphical abstract

- Additional sections at the end of the manuscript, include a Data Sharing Statement

- Impact statement

We believe that these revisions have significantly improved the quality of our manuscript and addressed all of the concerns raised by the reviewers.

We would like to thank you and the reviewers for your valuable feedback and guidance throughout the review process. We hope that our revised manuscript is now suitable for publication in your journal.

Sincerely,

Ana Carolina Florence.